# No-Touch Automated Disinfection System Based on Hydrogen Peroxide and Ethyl Alcohol Aerosols for Use in Healthcare Environments

**DOI:** 10.3390/ijerph19084868

**Published:** 2022-04-17

**Authors:** Francesco Triggiano, Giuseppina Caggiano, Marco Lopuzzo, Giusy Diella, Francesca Apollonio, Fabrizio Fasano, Maria Teresa Montagna

**Affiliations:** 1Department of Biomedical Science and Human Oncology, University of Bari Aldo Moro, Piazza G. Cesare 11, 70124 Bari, Italy; francesco.triggiano@uniba.it (F.T.); marco.lopuzzo@uniba.it (M.L.); 2Interdisciplinary Department of Medicine, Hygiene Section, University of Bari Aldo Moro, Piazza G. Cesare 11, 70124 Bari, Italy; giusy.diella@uniba.it (G.D.); francesca.apollonio@uniba.it (F.A.); f.fasano@regione.puglia.it (F.F.); mariateresa.montagna@uniba.it (M.T.M.)

**Keywords:** no-touch automated disinfection, hospital-acquired infections, hydrogen peroxide, ethyl alcohol, healthcare environment

## Abstract

Healthcare-related infections are sustained by various bacteria and fungi. In recent years, various technologies have emerged for the sanitation of healthcare-related environments. This study evaluated the effectiveness of a no-touch disinfection system that aerosolizes 5% hydrogen peroxide and 10% ethyl alcohol. After selecting an environment, the Total Bacterial Count and the Total Fungal Count in the air and on a surface of the room were determined to evaluate the effectiveness of the aerosolization system. In addition, sterile stainless-steel plates inoculated with *S. aureus*, *P. aeruginosa*, and *Aspergillus* spp. isolated from hospitalized patients and reference strains were used to evaluate the effectiveness of the system. For each organism, three plates were used: A (cleaned), B (not cleaned), and C (control). The A plates were treated with non-ionic surfactant and the aerosolization system, the B plates were subjected to the aerosolization system, and the plates C were positioned outside the room that was sanitized. Following sanitization, air and surface sampling was conducted, after which, swabs were processed for bacterial and fungal enumeration. The results showed that the air sanitization system had good efficacy for both bacteria and fungi in the air and on stainless-steel plates, particularly for the A plates.

## 1. Introduction

Hospital-acquired infections (HAIs) are a worldwide public health problem [1] and a significant cause of morbidity and mortality [2,3,4,5,6,7,8,9,10]. Even though standard decontamination procedures are insufficient, they still represent the main means of intervention to counter the circulation of pathogenic or potentially pathogenic microorganisms, and reduce infectious complications [11,12,13,14].

HAIs, which develop during care practices, are caused by various microorganisms, including multidrug-resistant (MDR) bacteria, such as Enterococci resistant to vancomycin (VRE), extended-spectrum β-lactamase-resistant *K. pneumoniae*, carbapenem-resistant *Pseudomonas aeruginosa*, *Acinetobacter* spp., and methicillin-resistant *Staphylococcus aureus* (MRSA) [8,15].

The main causes of HAIs are the so-called ESKAPE pathogens, which consist of the following organisms that exhibit multidrug resistance and virulence: *E. faecium*, *S. aureus*, *K. pneumoniae*, *A. baumannii*, *P. aeruginosa*, and *Enterobacter* species [16]. The reduction of these microorganisms, especially in nosocomial environments, is hindered by the presence of areas that are difficult to treat by simply cleaning and disinfecting surfaces [17]. Furthermore, for conventional methods to be effective, it is essential to use an appropriate disinfectant that is compatible with electro-medical equipment while ensuring the disinfectant is applied correctly and for the necessary contact time [17,18,19]. The use of “no-touch” automated room disinfection (NTD) systems eliminates or reduces dependency on operators, thereby improving the effectiveness of terminal disinfection [18,19]. 

The most common NTD systems currently used in healthcare facilities are hydrogen peroxide aerosol systems, H_2_O_2_ vapor systems, and ultraviolet C radiation systems [19]. It has been shown that the efficacy of these methods does not depend on the operator, which is a potential advantage over conventional cleaning [17]. Therefore, this study was conducted to evaluate the effectiveness of a NTD system for treatment of bacteria and fungi present in the air and on the surfaces of a hospital indoor environment.

## 2. Materials and Methods

The study was conducted in a room of known volume (25 m^3^) located in a healthcare facility. The doors to the room were closed, and there was no access allowed for the duration of the experiment and for 45 min after the treatment period. A no-touch disinfection system capable of delivering a pressurized aerosol solution of 5% hydrogen peroxide and 10% ethyl alcohol (SUPRASPOR—GIOCHEMICA s.r.l Monteforte d’Alpone (VR)—Italy—hereinafter referred to as HPEA) was used to treat the room. The aerosolized particles were <1 micron, and were introduced into a casing via a one-way nozzle. After exposure, the aerosols were allowed to decompose naturally.

Sampling was conducted in two phases. In the first phase, air sampling (total volume aspirated = 1000 L) was conducted using an active sampling system that drew air onto solid substrate (SAS, Surface Air System; PBI International, Milan, Italy) to determine the total bacterial count (TBC) and total fungal count (TFC) present before treating the room. At the same time, surface sampling (100 cm^2^) of a desk was conducted using sterile swabs (Easy Surface Checking—Neutralizing Rinse Solution; Liofilchem Srl, Roseto degli Abruzzi, Italy). After sampling, the swabs were placed into tubes containing 10 mL of transport medium in accordance with the recommendations of UNI EN 17141:2021 [20].

In the second phase, the effectiveness of the aerosolization system was evaluated. To accomplish this, nine sterile stainless-steel plates (42 cm^2^) were inoculated with nosocomial microorganisms (three replicate plates containing *Staphylococcus aureus*, *Pseudomonas aeruginosa*, or *Aspergillus flavus*) at a concentration of 1 McFarland (McF) (300 × 10^6^ colony forming units/mL) as described below. In parallel, standard strains of *Staphylococcus aureus* (NCTC 6571) and *Pseudomonas aeruginosa* (NCTC 10662) from the National Collection of Type Cultures (London, UK) and *Aspergillus brasiliensis* (NCPF 2275) from the National Collection of Pathogenic Fungi (Bristol, UK) were used as controls.

The top of a desk in the room to be disinfected was covered with a sterile non-woven fabric cloth, after which, the stainless-steel plates were positioned on top of the cloth. The plates were then inoculated by dipping a sterile swab into a suspension of one the aforementioned microorganisms, and uniformly rubbing the entire surface. Subsequently, the plates were allowed to dry for 10 min at room temperature. Next, plate A for each organism (hereinafter referred to as cleaned plate A) was sprayed with a non-ionic surfactant and swabbed with sterile gauze to remove excess detergent. Plate B for each organism (hereinafter referred to as no-cleaned plate B) was treated by the aerosolization system only. Plate C (hereinafter referred to as control plate C) was positioned outside the room to be sanitized, and not subjected to any treatment.

In the third phase, the sanitization of the environment started by using the system to apply HPEA. The volume of the room (25 m^3^) and the duration of the sanitization process (1 min and 8 s) were set on the sanitizing device. The disinfectant aerosolization time is established by the instrument according to the volume of the room to be sanitized. The system was programmed to provide a 30 s delay to allow the operators to leave the room, after which air diffusion of HPEA began. Following treatment, the room remained closed for 45 min. Air sampling and bench sampling was repeated for post-aerosolization TBC and TFC assessment.

The inoculated stainless-steel plates were sampled by wiping with a swab containing 10 mL of neutralizing transport medium. To determine the TBC, 1 mL of transport medium of each suspension was mixed and plated on Plate Count Agar (Microbiol Snc, Cagliari, Italy), incubated at 30 ± 1 °C, and checked daily for 72 ± 3 h (UNI EN ISO 4833-1:2013) [21]. To determine the TFC, 1 mL of transport medium of each dilution was mixed in duplicate with Sabouraud + Chloramphenicol (0.5 g/L) (Liofilchem, Roseto degli Abruzzi, Italy), incubated at 25 ± 2 °C, and checked for 5 days (NF V08-059:2002) [22]. All samples were plated in duplicate. After incubation, the presence of the colonies was expressed as colony-forming units (cfu) per cm^2^ (cfu/cm^2^).

The protocol described refers to studies already published with minor modifications [17,23,24].

### Statistical Analysis

To understand if there were statistically significant differences in the Total Bacterial Count (TBC) and Total Fungal Count (TFC) trend between cleaned plate A, no-cleaned plate B, and control plate C, the non-parametric Wilcoxon signed rank test with continuity correction was applied. A statistically significant result was considered with *p*-values < 0.05. We used R version 3.6.3 in the statistical analysis (The R Project for Statistical Computing, Vienna, Austria).

## 3. Results

The TBC and TFC of the air before and after disinfection by HPEA are shown in Table 1. The results revealed complete removal of the airborne bacterial load, and a large reduction of the fungal load. Additionally, only *A. fumigatus* was isolated, whereas no *A. niger* was found in the post-reclamation phase. The results of the desk sampling before and after disinfection demonstrated the absence (<1 cfu/cm^2^) of bacteria and fungi.

Table 2 shows the results for the stainless-steel plates (cleaned plate A, no-cleaned plate B, and control plate C) inoculated with nosocomial and reference strains. The load of the *S. aureus* nosocomial strain was 7100 cfu/cm^2^ on plate C, 5400 cfu/cm^2^ on plate B, and 2600 cfu/cm^2^ on plate A. The load of *P. aeruginosa* nosocomial strain was 1300 cfu/cm^2^ on plate C, 550 cfu/cm^2^ on plate B, and 200 cfu/cm^2^ on plate A. The load of *A. flavus* nosocomial strain was 76 cfu/cm^2^ on plate C, 4 cfu/cm^2^ on plate B, and <1 cfu/cm^2^ on plate A. 

All standard reference strains showed similar changes in concentration as the respective nosocomial strains (decreasing from plate C to B, and then A), with the exception of *A. brasiliensis*, which was not detected on plate B or A.

A Wilcoxon signed rank test with continuity correction led to statistically significant results by comparing both the trend of the Total Bacterial Count (TBC) and Total Fungal Count (TFC) (cfu/cm^2^) between cleaned plate A and control plate C (V = 21, *p*-value = 0.03); between no-cleaned plate B and control plate C (V = 21, *p*-value = 0.03); and between cleaned plate A and no-cleaned plate B (V = 15, *p*-value = 0.05). 

## 4. Discussion

Air and surfaces represent important sources of transmission of pathogenic microorganisms in hospital settings. Microorganisms normally present in the air are generally harmless to healthy people, but the pathological conditions of the patient, and increased duration of exposure can lead to a greater chance of infection [25,26]. Moreover, healthcare environments are characterized by various critical environmental conditions. Therefore, the spread, survival, and persistence of microbial communities are important risk factors for patient and medical staff health [27,28,29].

The COVID-19 pandemic has seriously endangered the world health system. In particular, hospitals have had to deal with unusually high frequencies of patients in emergency departments, particularly in intensive care units [30]. This pandemic has highlighted the importance of paying greater attention to the transmission of airborne diseases, hand washing, and the development of more effective methods of environmental disinfection aimed at reducing the concentration and/or viability of airborne microorganisms, including viruses.

Conventional cleaning and disinfection performed by human operators using liquid detergents or disinfectants rarely eradicate pathogens from the environment. Moreover, the product used, and failure to properly follow manufacturer guidelines (e.g., appropriate product selection and formulation, distribution, and contact time), can affect the expected results [19].

Our study evaluated a no-touch disinfection system based on the aerosolization and application (for 1 min and 8 s) of a solution comprising 5% hydrogen peroxide and 10% ethyl alcohol (HPEA). This system led to a large decrease in the airborne bacterial load. In addition, a good reduction of *Aspergillus fumigatus*, and complete disappearance of *Aspergillus niger*, which were present in the air before the experiment, was also observed. On the desktop, the presence of microorganisms was not detected before disinfection, probably because of the daily cleaning that is carried out routinely.

Evaluation of experimentally inoculated stainless-steel plates revealed a statistically significant reduction in bacterial and fungal load (plate A vs. plate C *p* = 0.03; plate B vs. plate C *p* = 0.03; plate A vs. plate C *p* = 0.05), with greater reductions being observed on plates that were cleaned and then subjected to aerosol treatment, regardless of the strains tested. Similar results were observed for the fungi that were evaluated. These data show that disinfection with HPAE results in satisfactory levels of surface sanitation, especially if associated with manual cleaning. Some studies have shown that aerosolized H_2_O_2_ systems also act on hospital surfaces inoculated with more difficult-to-manage bacteria, such as *Clostridium difficile* and MRSA [31,32]. Rios-Castillo et al. [23] showed that formulations of hydrogen peroxide mixed with other disinfectant products have acceptable bactericidal efficacy on stainless-steel surfaces, thereby enabling the total concentration of hydrogen peroxide and common disinfectants to be reduced.

It is important to note that this was a preliminary in vitro study, and the concentration of microorganisms used on stainless-steel surfaces was higher than that generally found on common hospital surfaces. This was chosen to simulate the worst-case scenarios for a hospital. We plan to conduct future evaluations of the tested system in larger and more heavily-furnished rooms.

It is also necessary to highlight that, according to the instrument’s technical data sheet, this system poses no risk to the operators, is extremely safe to use on any type of material, and requires very short action times. Introducing this equipment into common disinfection procedures could help decrease the number of HAIs, resulting in reduced consumption of antibiotics, and decreased pharmaceutical costs for healthcare facilities.

## 5. Conclusions

This disinfection system based on the use of a solution of 5% hydrogen peroxide and 10% ethyl alcohol is not intended to completely replace the manual cleaning and disinfection of surfaces. Rather, it is a complementary process capable of improving environmental quality, and reducing the risk of contamination, especially in areas that are difficult to disinfect or access.

Air sampling revealed good efficacy of the system toward both bacteria and fungi. Additionally, sampling of stainless-steel plates revealed reductions in TBC and TFC, even in the absence of cleaning, despite the high concentrations of organisms that were treated. Finally, combined cleaning and aerosol treatment (plate A) produced excellent results in terms of reducing the microbial load. Further studies are needed to evaluate the effectiveness of the system for treatment of other materials (e.g., plastic, glass).

## Figures and Tables

**Table 1 ijerph-19-04868-t001:** Airborne Total Bacterial Count (TBC) and Total Fungal Count (TFC) pre-/post-disinfection by hydrogen peroxide (5%) and ethyl alcohol (10%).

	TBC	TFC
Pre-disinfection	110 cfu/m^3^	83 cfu /m^3 (^*^)^
Post-disinfection	<1 cfu/m^3^	6 cfu/m^3 (°)^

(*) *Aspergillus fumigatus* + *Aspergillus niger*. (°) *Aspergillus fumigatus*.

**Table 2 ijerph-19-04868-t002:** TBC and TFC (cfu/cm^2^) of stainless-steel plates inoculated with nosocomial and standard reference strains.

Strains	Cleaned Plate A (cfu/cm^2^)	No-Cleaned Plate B (cfu/cm^2^)	Control Plate C (cfu/cm^2^)
Nosocomial *S. aureus*	2600	5400	7100
NCTC 6571—*S. aureus*	120	2800	3500
Nosocomial *P. aeruginosa*	200	550	1300
NCTC 10662—*P. aeruginosa*	1200	2300	3300
Nosocomial *A. flavus*	<1	4	76
NCPF 2275—*A. brasiliensis*	<1	<1	60

## Data Availability

Not applicable.

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
