# Peer review of "No-Touch Automated Disinfection System Based on Hydrogen Peroxide and Ethyl Alcohol Aerosols for Use in Healthcare Environments"

_ijerph, 2022, doi:10.3390/ijerph19084868_

Round 1
Reviewer 1 Report
This study evaluates the effectiveness of a no-touch disinfection system using the total bacterial count and total fungal count in the air and on a surface of the room located in a healthcare facility. The idea is clearly presented and easy to follow. The paper is well-organized.
It looks like citation eight that was published in this journal had done more extensive experiments such as in 50 surfaces in 10 hospital rooms, although they used microbiological assay and an ATP bioluminescence assay to assess the contamination of the surfaces, are there any advantages of yours compared to it?
Author Response
Reviewer 1
We thank the Reviewer for the comments and observations.
This study evaluates the effectiveness of a no-touch disinfection system using the total bacterial count and total fungal count in the air and on a surface of the room located in a healthcare facility. The idea is clearly presented and easy to follow. The paper is well-organized.
It looks like citation eight that was published in this journal had done more extensive experiments such as in 50 surfaces in 10 hospital rooms, although they used microbiological assay and an ATP bioluminescence assay to assess the contamination of the surfaces, are there any advantages of yours compared to it?
Our study aimed to evaluate the aerosolization system in the room in question and to evaluate also in vitro the effectiveness on the main microorganisms responsible for HAI in healthcare environments. In fact, in the manuscript we reported that ours was an experimental in vitro study and the concentration of microorganisms used on stainless steel surfaces was higher than that generally found on common hospital surfaces. This was done to simulate the worst-case scenarios for a hospital. We plan to conduct future evaluations of the tested system in larger and more heavily furnished rooms.

Reviewer 2 Report
This manuscript presents a very relevant topic for public health. It is very clear and well-written. I just have a few questions for authors to clarify:
- In the methods section, could you elaborate why the duration of the sanitation process was set up for 1 minute and 8 seconds? Did you tried with other times before this experiment?
- In the discussion section, authors mention the COVID-19 pandemic to highlight the importance of disinfection, especially in healthcare environments, and with hospitalized patients with other comorbidities. Does this disinfection system could be used for COVID-19?
Only at healthcare facilities or could it be used in other places with high amounts of populations? If so, could you elaborate a bit more on that?
Author Response
We thank the Reviewer for the comments and observations.
This manuscript presents a very relevant topic for public health. It is very clear and well-written. I just have a few questions for authors to clarify:
- In the methods section, could you elaborate why the duration of the sanitation process was set up for 1 minute and 8 seconds? Did you tried with other times before this experiment?
The disinfectant aerosolization time is established by the instrument according to the volume of the room to be sanitized. We have added this sentence in the manuscript (lines 96-97). We haven't tried other times.
- In the discussion section, authors mention the COVID-19 pandemic to highlight the importance of disinfection, especially in healthcare environments, and with hospitalized patients with other comorbidities. Does this disinfection system could be used for COVID-19?
According to the manufacturer, this sanitization system also acts against viruses, but we have not carried out experimental tests to evaluate the effectiveness against SARS-CoV-2.
Only at healthcare facilities or could it be used in other places with high amounts of populations? If so, could you elaborate a bit more on that?
Yes, this sanitization system can be used in any environment. The experimentation with stainless steel plates was aimed at evaluating the effectiveness of the system against bacteria and fungi that are found mostly in healthcare facilities, but also, in lower concentrations in environments such as gyms, swimming pools, theaters, schools, etc.

Reviewer 3 Report
ijerph-1676455
Type of manuscript: Article
Title: No-Touch Automated Disinfection System based on hydrogen peroxide and ethyl alcohol aerosols for use in healthcare environments
Authors: Francesco Triggiano, Giuseppina Caggiano, Marco Lopuzzo, Giusy Diella, Francesca Apollonio, Fabrizio Fasano, Maria Teresa Montagna
The paper presented for review describes research on the use of the new system for non-contact disinfection of hospital rooms and surfaces with a mixture of hydrogen peroxide and ethyl alcohol. Experiments involving the removal of common bacteria and fungi from the air and from metal plates have been described. The presented results are promising, however, an in-depth analysis of the presented issue is necessary.
- Many of the experimental conditions were set arbitrarily without optimization or literature analysis.
- The work does not deal with an important aspect related to the chemical analysis of air and the residence time of hydrogen peroxide and ethyl alcohol in it.
- It is not clear why these concentrations and the proportion of disinfectants and the duration of action of the mixture were chosen.
- More experiments need to be carried out and the mean content of microorganisms in the air and on surfaces as well as standard deviations should be presented.
Author Response
We thank the Reviewer for the comments and observations.
The paper presented for review describes research on the use of the new system for non-contact disinfection of hospital rooms and surfaces with a mixture of hydrogen peroxide and ethyl alcohol. Experiments involving the removal of common bacteria and fungi from the air and from metal plates have been described. The presented results are promising, however, an in-depth analysis of the presented issue is necessary.
- Many of the experimental conditions were set arbitrarily without optimization or literature analysis.
For this experimentation we referred to previously published studies making some changes having used an instrument that aerosolises the disinfectant. We have included this information in the manuscript (lines 111-112).
- The work does not deal with an important aspect related to the chemical analysis of air and the residence time of hydrogen peroxide and ethyl alcohol in it.
Hydrogen peroxide and ethyl alcohol are compounds that evaporate easily in the environment. Furthermore, a period of non-use of the room of 45 minutes was guaranteed to allow the natural decomposition of the airborne substances.
- It is not clear why these concentrations and the proportion of disinfectants and the duration of action of the mixture were chosen.
The non-contact disinfection system is capable of delivering a pressurized aerosol solution of 5% hydrogen peroxide and 10% ethyl alcohol. This product is ready to use and does not require any dilution according to the instrument's technical data sheet.
- More experiments need to be carried out and the mean content of microorganisms in the air and on surfaces as well as standard deviations should be presented.
Yes, as already reported in the manuscript this is an experimental study that we would like to expand.

Reviewer 4 Report
The authors present an interesting study using aerosol disinfectants for clinically important infections. The manuscript is very well written. I have a few minor comments and queries as follows.
Statistical analysis descriptions are typically within the methods sections.
What statistical tests were actually carried out? These need to presented in the results section.
There are areas of text in the discussion that describe a significant result, but there is not enough stats provided in the results to support all of these statements.
Consider rephrasing line 154 to 'influencing both the health of patients and medical staff'
Insert P value after 'significant' in line 174
Check the grammar in the conclusion. There are several commas missing
Author Response
We thank the reviewer for the comments and observations.
The authors present an interesting study using aerosol disinfectants for clinically important infections. The manuscript is very well written. I have a few minor comments and queries as follows.
Statistical analysis descriptions are typically within the methods sections.
We have modified. Now, Statistical analysis paragraph is in the methods sections.
What statistical tests were actually carried out? These need to presented in the results section.
In the results section (lines 145-149) we reported the results relating to the statistical analysis conducted with the method used, also described in the methods section.
There are areas of text in the discussion that describe a significant result, but there is not enough stats provided in the results to support all of these statements.
Although with a small number of microorganisms tested on the stainless steel plates, we carried out a stratistic analysis between the bacteria tested (nosocomial strains and control strains) and between the tested molds (nosocomial strain and control strain). The materials and methods section shows the details relating to the statistical survey conducted only on the in vitro study.
Consider rephrasing line 154 to 'influencing both the health of patients and medical staff'
We have modified this sentence.
Insert P value after 'significant' in line 174
We have added this information in the manuscript (lines 179-180).
Check the grammar in the conclusion. There are several commas missing
We thanks.
Round 2
Reviewer 3 Report
The comments presented by the authors clarified the previous doubts and the corrections made increased the quality of the work. Therefore, the article may be accepted for publication.